# Cytomegalovirus drives Vδ1+ γδ T cell expansion and clonality in common variable immunodeficiency

Samantha Chan [1,2,3,4], Benjamin Morgan[1], Michelle K. Yong [5,6,7], Mai Margetts [1], Anthony J. Farchione [1,2], Erin C. Lucas [1], Jack Godsell[3,8], Nhi Ai Giang [3], Charlotte A. Slade[1,2,3], Anouk von Borstel [9,10], Vanessa L. Bryant[1,2,3,11] & Lauren J. Howson [1,2,11] ✉

The function and phenotype of γδ T cells in the context of common variable immunodeficiency (CVID) has not been explored. CVID is a primary immunodeficiency disorder characterized by impaired antibody responses resulting in increased susceptibility to infections. γδ T cells are a subset of unconventional T cells that play crucial roles in host defence against infections. In this study, we aim to determine the roles and functions of γδ T cells in CVID. We observe a higher frequency of Vδ1+ γδ T cells compared to healthy controls, particularly in older patients. We also find a higher proportion of effector-memory Vδ1+ γδ T cells and a more clonal T cell receptor (TCR) repertoire in CVID. The most significant driver of the Vδ1+ γδ T cell expansion and phenotype in CVID patients is persistent cytomegalovirus (CMV) viremia. These findings provide valuable insights into γδ T cell biology and their contribution to immune defence in CVID.

Common variable immunodeficiency (CVID) is a primary immunodeficiency disorder clinically defined as a diagnosis of exclusion for individuals with low serum immunoglobulin (Ig)G, impaired vaccine responses, and recurrent sinopulmonary infections[1]. The cause for CVID is heterogeneous, with approximately 20% of patients having a confirmed genetic diagnosis[2,3]. The majority of CVID patients also develop immune dysregulation, with complications such as autoimmunity or inflammation often requiring immunosuppression[4], leaving these individuals vulnerable to secondary infections[5].

Cytomegalovirus (CMV) is an opportunistic beta-herpes virus that can cause serious disease in the immunocompromised. CMV has a seroprevalence of 40–90% and is typically an asymptomatic infection followed by latency in healthy individuals[6,7]. In an immunocompromised setting, such as following solid organ or hematopoietic stem cell transplantation (HSCT), CMV can become reactivated, persist, and in some cases lead to CMV disease. This is characterized by tissue invasive manifestations, such as enteritis, esophagitis, or pneumonitis[8].

γδ T cells mount a robust immune response to CMV infection[9,10]. As an unconventional T cell population, they are not typically restricted by classical major histocompatibility complex (MHC) for ligand recognition[11]. γδ T cells make up to 10% of circulating T cells and respond rapidly to microbial infection and cancer[12–14]. They are predominately composed of two subsets defined by their T cell receptor (TCR)δ chain usage. The Vδ2+ (predominantly paired with Vγ9 chain)

[1]Immunology Division, Walter & Eliza Hall Institute of Medical Research, Melbourne, VIC, Australia. [2]Department of Medical Biology, The University of Melbourne, Melbourne, VIC, Australia. [3]Department of Clinical Immunology & Allergy, Royal Melbourne Hospital, Melbourne, VIC, Australia. [4]Department of Medicine, The University of Melbourne, Melbourne, VIC, Australia. [5]Victorian Infectious Diseases Service, Royal Melbourne Hospital, Melbourne, VIC, Australia. [6]Sir Peter MacCallum Department of Oncology, The University of Melbourne, Melbourne, VIC, Australia. [7]National Centre for Infections in Cancer, Peter MacCallum Cancer Centre, Melbourne, VIC, Australia. [8]Department of Infectious Diseases, Austin Hospital, Heidelberg, VIC, Australia. [9]Infection and Immunity Program and Department of Biochemistry and Molecular Biology, Biomedicine Discovery Institute, Monash University, Clayton, VIC, Australia. [10]Present address: Department of Immunology, Central Clinical School, Monash University, Melbourne, VIC, Australia. [11]These authors contributed equally: Vanessa L. Bryant, Lauren J. Howson. ✉e-mail: howson.l@wehi.edu.au

subset is the most abundant in circulation and is considered innate-like as they uniformly recognize and rapidly respond to phosphoantigens, such as (E)−4-hydroxy-3-methyl-but-2-enyl pyrophosphate (HMBPP) produced by a range of bacteria and parasites[15]. The Vδ1+ subset is typically abundant at mucosal sites and is considered more adaptive-like as they can transition from naïve to effector cells in various infectious diseases and have a predominately private TCR repertoire recognizing a range of largely unknown antigens[13,16,17]. It is the Vδ1+ γδ T cell subset that reportedly undergo clonotypic expansion in response to CMV reactivation post-transplant[18]. Despite this knowledge, the response of γδ T cells to CMV viremia in the context of primary immunodeficiency remains poorly understood.

The T cell compartment in CVID patients has been observed as having reduced thymic output, CD4 T cell lymphopenia, and increased T cell activation and exhaustion[19]. However, little is known about γδ T cells in the context of CVID. A study has previously reported an inverted Vδ1/Vδ2 ratio in circulating γδ T cells in CVID patients, which did not resolve with intravenous Ig (IVIg) treatment[20]. However, the Vδ1/Vδ2 ratio also becomes inverted with age, making it an important factor that needs to be considered when studying γδ T cell frequencies in disease settings[21]. Examining γδ T cells in individuals with impaired humoral immunity, such as those with CVID, provides an opportunity to understand whether normal γδ T cell function requires intact B cell immunity. Previous studies have suggested that B cells can depend on γδ T cell help, but it is unknown whether the γδ T cells rely on B cells for any aspect of their functionality[22–24]. Studying CVID patients also enables examination of how γδ T cell subsets respond in an individual that is frequently challenged with infections and inflammatory complications. Furthermore, in CVID patients with persistent CMV viremia, it provides insight into CMV-specific γδ T cell responses in the context of primary immunodeficiency.

We conduct a study of patients diagnosed with CVID where we phenotype and functionally characterize their γδ T cell subsets. We include CVID patients diagnosed with CMV viremia and analyze the impact this had on their γδ T cells. Our results demonstrate that individuals across all ages with CVID have γδ T cells that mirror the frequency and clonotypic expansions observed in healthy aging/older individuals. The γδ T cell population is not impaired due to the lack of B cell immunity in these individuals. Furthermore, CMV viremia in CVID patients drives the Vδ1+ clonotypic expansion and activation profile of these cells, demonstrating that anti-CMV γδ T cell immunity is intact in the context of CVID.

## Results

### Vδ1+ γδ T cells are expanded, activated, and functional in CVID
To characterize the γδ T cell population in patients with CVID, the number of circulating γδ T cells in peripheral blood samples of patients with CVID was examined and compared to healthy individuals (See Table 1 for cohort summary, Supplementary Fig. 1 for gating strategy and Supplementary Table 1 for CVID cohort details). There was a significantly higher number of circulating γδ T cells in CVID patients (119 ± 37 cells/μL) compared to healthy individuals (46 ± 7.2 cells/μL) (Fig. 1a, see Supplementary Table 2 for detailed T cell subset counts and frequencies). We observed a significantly higher frequency of Vδ1+ γδ T cells as a proportion of total T cells in CVID patients (3.9 ± 1.5%) compared to healthy individuals (0.7 ± 0.1%) (Fig. 1b). This resulted in a significantly higher Vδ1/Vδ2 ratio in CVID (Fig. 1c); however, the cell counts for the γδ subsets were not significantly different between healthy and CVID (Supplementary Fig. 2a). There were also no significant differences in the circulating frequencies of Vδ2+ or Vδ1−Vδ2− γδ T cells (Fig. 1b).

We next evaluated the effector status of circulating γδ T cells by examining the proportion of cells in an effector-like state in CVID patients compared to healthy individuals. We found that the frequency of effector-like CD27−CX3CR1+ Vδ1+ γδ T cells was significantly elevated (41 ± 18%) in CVID patients compared to healthy individuals (21 ± 2%) (Fig. 1d). There was also a significantly higher proportion of effector-like CD27−CD28− Vδ2+ γδ T cells (38 ± 4.3%) compared to healthy individuals (13 ± 1.2%) (Fig. 1d). We also examined the cytotoxic potential of circulating γδ T cell subsets and found significantly higher proportions of granzyme B+ Vδ1+ (54 ± 7.3%) and Vδ2+ (63 ± 6%) γδ T cells in CVID patients compared to healthy individuals (30 ± 3.9% and 35 ± 3.8%, respectively) (Fig. 1e). This also corresponded with a significantly higher proportion of perforin+ Vδ1+ (49 ± 8%) and Vδ2+ (55 ± 7%) γδ T cells in CVID patients compared to healthy individuals (29 ± 3% and 33 ± 3%, respectively) (Fig. 1f).

To assess the polyfunctionality of circulating γδ T cells in CVID, we stimulated peripheral blood mononuclear cells (PBMC) with a range of stimuli. We found that Vδ1+ and Vδ2+ γδ T cells from CVID patients exhibited intact pro-inflammatory effector functions that were comparable to the response observed in healthy donors (Fig. 1g and Supplementary Fig. 3). Thus, CVID patients' circulating γδ T cells maintain their function and cytotoxic potential and have a significant expansion of effector-like cells.

### Vδ1+ and Vδ2+ γδ T cell frequencies correlates with age
We investigated the impact of age on the frequency of circulating γδ T cells in CVID patients and healthy individuals. The total γδ T cell count (Fig. 2a) and frequency (Fig. 2b) did not significantly correlate with age in either healthy or CVID cohorts. However, the Vδ1+ subset frequency (as a proportion of total γδ T cells) exhibited a significant positive correlation (R² = 0.10) with age in healthy individuals and a stronger significant positive correlation (R² = 0.24) in CVID patients (Fig. 2c). Conversely, the Vδ2+ subset exhibited a significant negative correlation associated with age in both healthy individuals (R² = 0.14) and CVID patients (R² = 0.26) (Fig. 2d). This age-related correlation was not observed for the Vδ1−/Vδ2− subset (Supplementary Fig. 2b). We then investigated the impact of age on effector status of the γδ T cell subsets and found a significant positive correlation between CD27−CX3CR1+ Vδ1+ γδ T cells in both healthy (R² = 0.14) and CVID patients (R² = 0.20) (Fig. 2e) but not between CD27−CD28− Vδ2+ γδ T cells and age (Fig. 2f). We also examined the impact of sex, which had minimal effect on γδ subset frequencies (Supplementary Fig. 4). Multivariate analysis confirmed that age and CVID, but not sex, are both significantly associated with an increased Vδ1 frequency (Supplementary Table 3).

### CMV viremia is associated with a skewed γδ T cell phenotype
To understand the potential pathological drivers of the circulating γδ T cell expansion and effector phenotype we observed in CVID, we first grouped patients based on their non-infectious clinical complications and found no significant differences in γδ subset frequencies (Supplementary Fig. 5). We then examined their infectious history and

## Table 1 | Summary of cohort characteristics

| Characteristics | | CVID n = 23 | Healthy controls n = 50 |
|---|---|---|---|
| Age (median, range) | | 46, 19–75 | 46, 18–77 |
| Sex | Female | 16 (67%) | 26 (52%) |
| | Male | 7 (33%) | 24 (48%) |
| CMV serology | Positive | 0 | 15/50 |
| | Negative | 0 | 19/50 |
| | Not determined | 23/23ᵃ | 16/50 |
| CMV viremia | | 5/23 | ND |
| Clinical genomics | Tested | 17/23 | ND |
| | Diagnosed | 8/23 | ND |

*IGRT* immunoglobulin replacement therapy, *ND* not determined.
ᵃCMV serology not determined for CVID patients due to low IgG production/IGRT treatment confounding serology results.

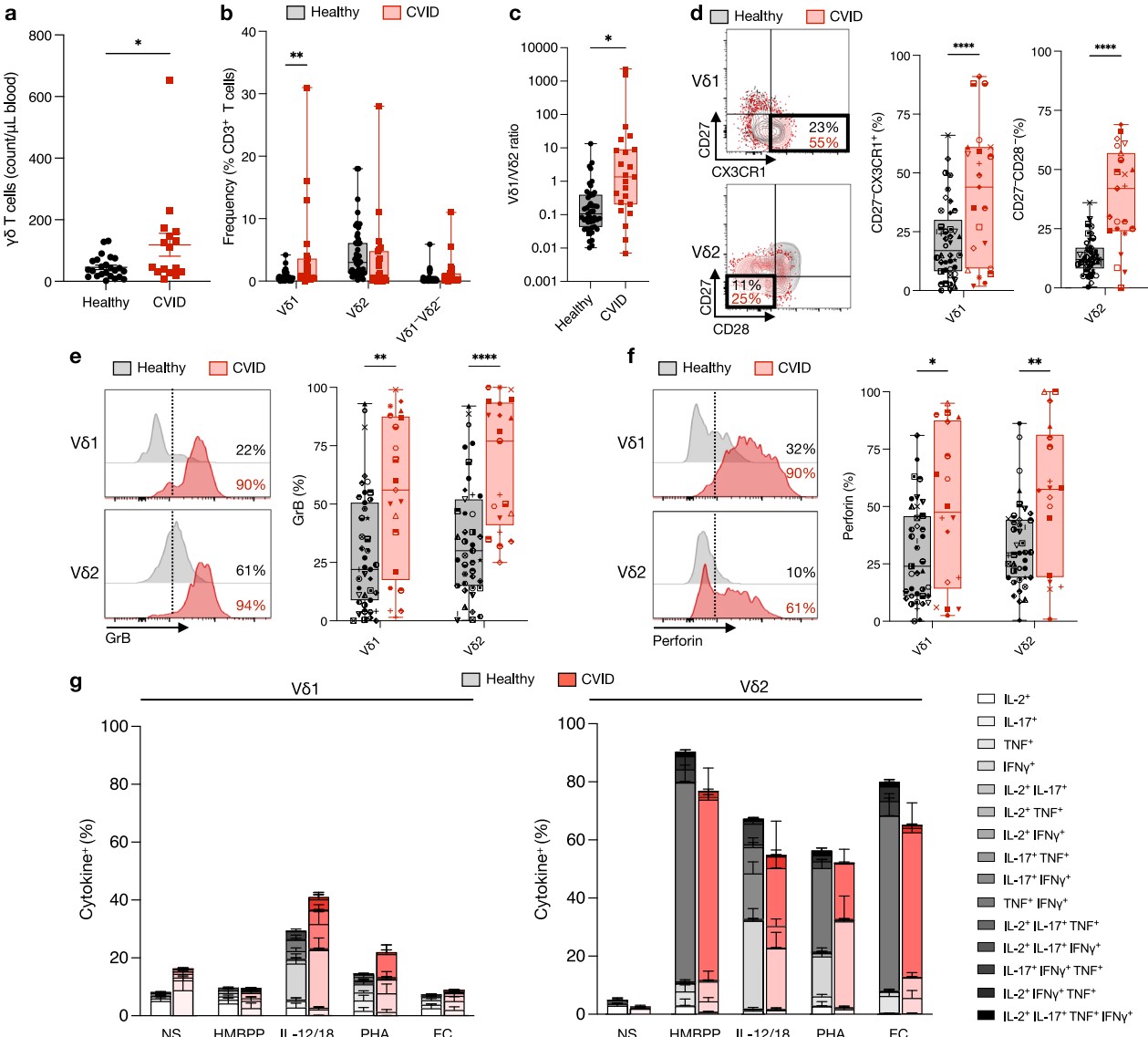

**Fig. 1 | Frequency, phenotype, and function of γδ T cells in CVID.** Peripheral blood mononuclear cells (PBMC) from healthy individuals and common variable immunodeficiency (CVID) patients were analyzed by flow cytometry (see gating strategy in Supplementary Fig. 1). **a** Graph of the circulating number of γδ T cells ($n = 24$ healthy, 17 CVID) *$P = 0.029$ and (**b**) Vδ1$^+$, Vδ2$^+$, and Vδ1$^-$/Vδ2$^-$ subsets as a frequency of total CD3$^+$ T cells ($n = 47$ healthy, 23 CVID) **$P = 0.004$. **c** Graph of the ratio of Vδ1/Vδ2 in healthy individuals and CVID patients ($n = 47$ healthy, 23 CVID) *$P = 0.040$. Plots and graphs show (**d**) the CD27$^-$ CX3CR1$^+$ Vδ1$^+$ and CD27$^-$ CD28$^-$ Vδ2$^+$ effector populations ($n = 44$ healthy, 23 CVID) **** $P < 0.0001$, **e** granzyme B$^+$ ($n = 43$ healthy, 21 CVID) **$P = 0.002$, ****$P < 0.0001$, and (**f**) perforin$^+$ ($n = 40$ healthy, 18 CVID) *$P = 0.011$, **$P = 0.005$ populations for Vδ1$^+$ and Vδ2$^+$ subsets. **g** Stacked graph shows Vδ1$^+$ and Vδ2$^+$ subsets polyfunctional cytokine response to various stimuli measured by intracellular staining for IL-2, TNF, IL-17 and IFNγ after 18 h of stimulation in the presence of brefeldin A (BFA) in healthy individuals ($n = 8$) and CVID patients ($n = 6$). For all graphs, each point represents an individual (unique symbols for each individual are shown in **d**–**f**). For bar graphs, bars represent the mean and error bars represent standard error of the mean (SEM). For box and whisker graphs, line is at median, box indicates upper and lower quartiles, and error bars are minimum and maximum values. Statistical significance was calculated using either two-tailed unpaired T test or two-way analysis of variance (ANOVA) and Sidak's multiple comparison test. EC *E. coli*, GrB granzyme B, HMBPP (E)−4-hydroxy-3-methyl-but-2-enyl pyrophosphate, IFNγ interferon gamma, IL interleukin, NS no stimulation, PHA phytohemagglutinin, TNF tissue necrosis factor.

identified five patients in our CVID cohort with persistent CMV viremia and/or CMV disease (CMV/CVID patient overview shown in Fig. 3, Supplementary Table 4 details full clinical history). CMV infection in healthy individuals is known to drive age-dependent Vδ1$^+$ γδ T cell expansion[25]. Thus, we separated individuals in our healthy cohort into CMV seronegative (CMV$^-$) and seropositive (CMV$^+$) and observed that the Vδ1$^+$ γδ T cells did have a higher frequency in CMV$^+$ individuals (which was also age-dependent), but this difference was not statistically significant (Supplementary Fig. 6).

CMV/CVID patients had no significant difference in total circulating numbers of γδ T cells compared to healthy and CVID cohorts (Fig. 4a) and only a significantly higher frequency of total γδ T cells compared to CMV$^-$ healthy individuals (Supplementary Fig. 7a). Despite this, we did observe a significant increase in the Vδ1$^+$ subset frequency (12 ± 5.5%) within the T cell compartment compared to CMV$^-$ and CMV$^+$ healthy individuals and CVID patients (Fig. 4b). The Vδ1/Vδ2 ratio was also significantly more inverted in CMV/CVID patients (778 ± 481) compared to CVID patients (2.9 ± 1.4) (Fig. 4c). As all other T cell counts were not significantly different across cohorts and largely within the normal range (Supplementary Fig. 7b), this suggests the expansion of the Vδ1$^+$ subset is at the expense of the Vδ2$^+$ subset. This correlation between increased Vδ1$^+$ cell frequency and

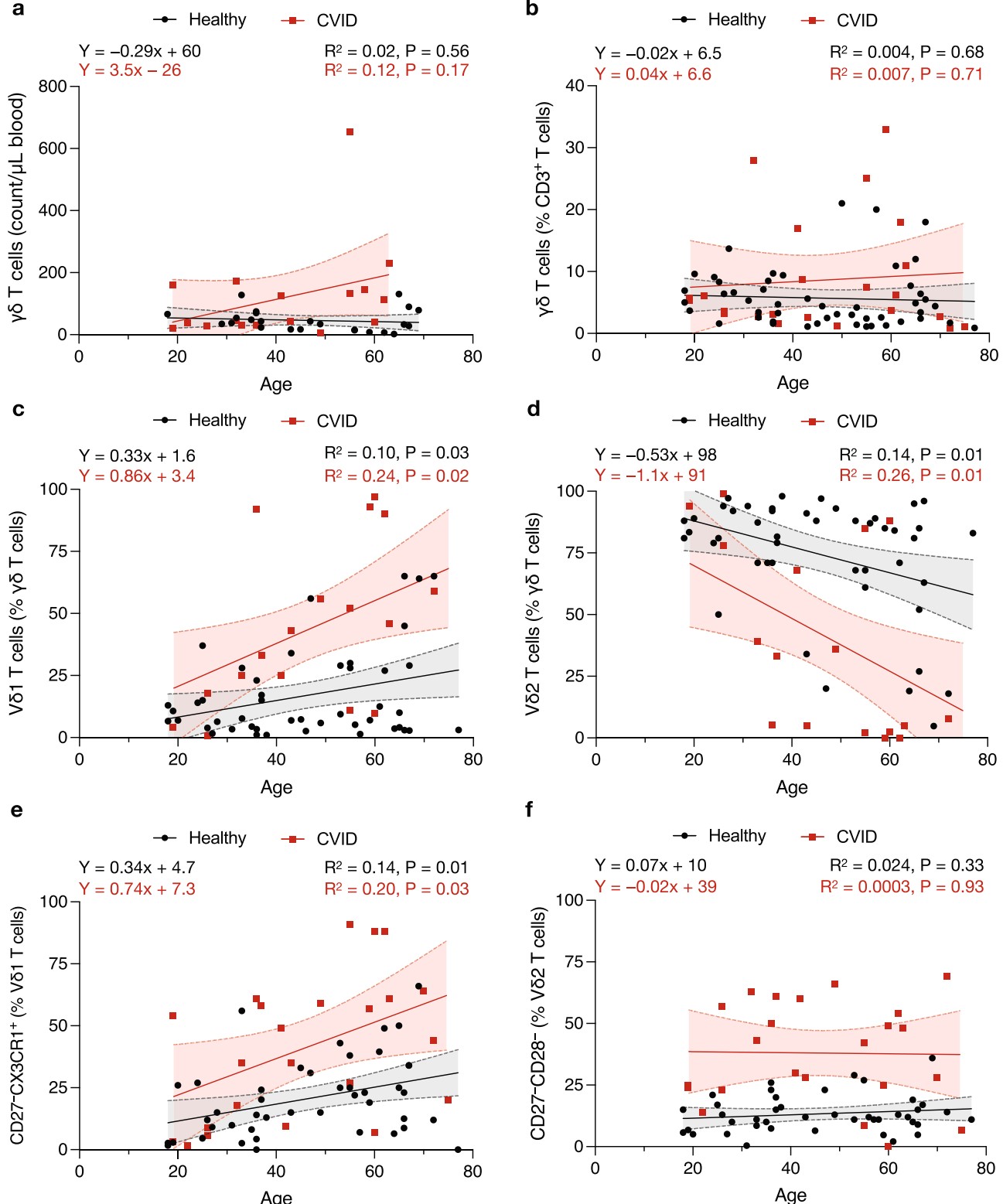

**Fig. 2 | Impact of age on γδ T cell frequency.** PBMCs from healthy individuals and CVID patients were analyzed by flow cytometry (see gating strategy in Supplementary Fig. 1). **a** Circulating γδ T cell number ($n = 24$ healthy, 17 CVID) and (**b**) frequency of total γδ T cells plotted against age ($n = 50$ healthy, 23 CVID). The proportion of γδ T cells that are (**c**) Vδ1+ and (**d**) Vδ2+ are plotted against age ($n = 46$ healthy, 23 CVID). The effector phenotype frequency for (**e**) Vδ1 and (**f**) Vδ2 are plotted against age ($n = 42$ healthy, 23 CVID). Line represents simple linear regression with 95% confidence interval of the best fit line shown as dashed line and shaded area. Statistical significance was calculated using simple linear regression (test for non-zero slope) with significance determined when $P < 0.05$ and $R^2$ indicates goodness of fit.

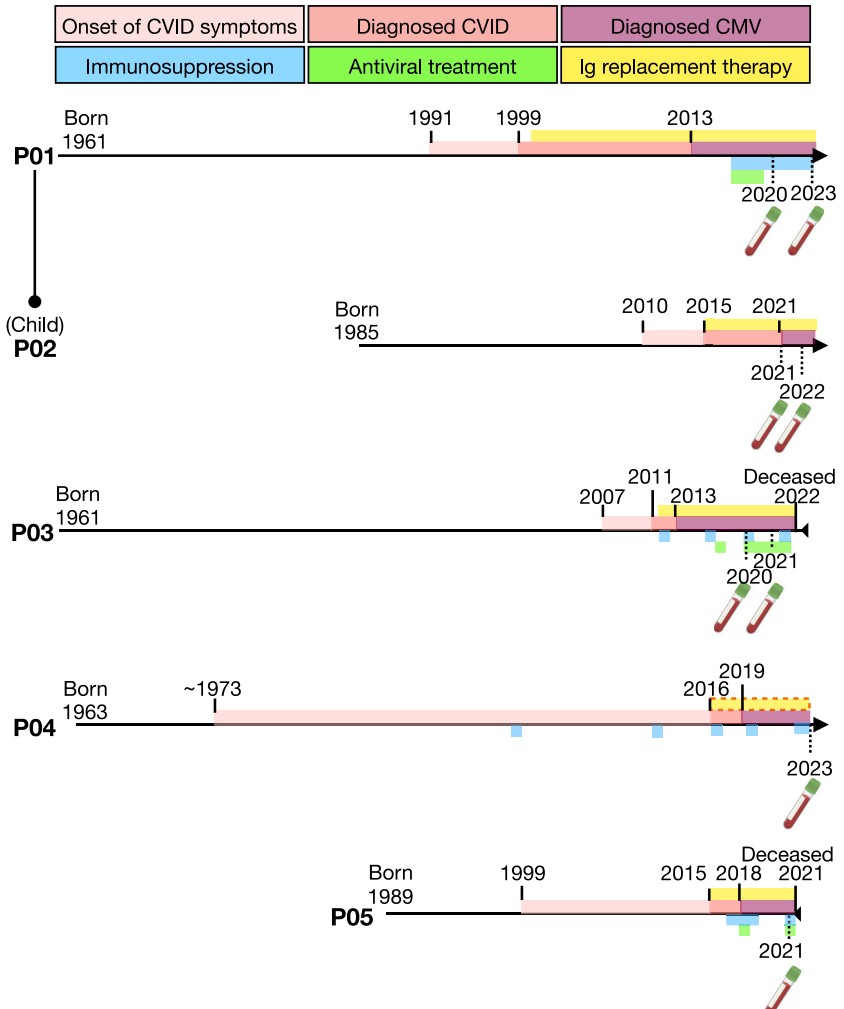

**Fig. 3 | Clinical and sampling overview for patients with CMV viremia.** Graphical timeline for five CVID patients diagnosed with cytomegalovirus viremia (CMV/CVID). The relationship between two of the patients (mother and child) is shown. Birth (and death, if applicable) year is shown. Year and timeframes (as a colored bar) are shown for onset of CVID symptoms, CVID diagnosis, and CMV viremia diagnosis (via PCR). Colored bars are also used to represent the time period for which patients are receiving: Ig replacement therapy (orange dashed outline indicates intermittent use), immunosuppression, or antiviral treatment for CMV. Time at which blood samples were taken for analysis is represented by the blood collection tube. Ig, immunoglobulin.

CMV/CVID was confirmed as significant by multivariate analysis (Supplementary Table 3).

The CMV/CVID patients also had significantly higher proportion of effector CD27$^-$CX3CR1$^+$ Vδ1$^+$ and CD27$^-$CD28$^-$ Vδ2$^+$ γδ T cells (Supplementary Fig. 7c) and cytotoxic (granzyme B$^+$) Vδ1$^+$ and Vδ2$^+$ cells (Supplementary Fig. 7d) compared to CMV$^-$ and CMV$^+$ healthy individuals. We also confirmed that the Vδ1$^+$ γδ T cells still maintained a polyfunctional cytokine response to stimulation comparable to healthy and CVID cohorts (Supplementary Fig. 7e).

We next investigated whether the γδ T cell subsets had altered coreceptor expression, a hallmark of chronic inflammation[26]. γδ T cells are typically (>70%) CD4$^-$/CD8$^-$ in circulation[27], the remaining being CD8$^+$ and <1% being CD4$^+$. We found a significantly higher proportion of CD8$^+$ Vδ1$^+$ γδ T cells in CMV/CVID patients (64 ± 7.9%) compared to CVID patients (34 ± 5.4%), and both CMV$^-$ and CMV$^+$ healthy individuals (16 ± 3.0% and 24 ± 4.4%, respectively) (Fig. 4d). We observed a small increase in CD8$^+$ Vδ2$^+$ γδ T cells in CVID patients compared to healthy individuals, but there was no significant difference when compared to CMV/CVID patients (Supplementary Fig. 7f).

We have previously reported that a higher proportion of patients with CMV viremia/disease have a monogenic cause for their CVID[8]. In the present study, four of the five CMV/CVID patients had a monogenic

cause for CVID. To address whether this is a confounding factor, we examined the circulating γδ T cell subset frequency in individuals with and without a genetic diagnosis and found that patients with a monogenic cause for CVID did not have a significant difference in their Vδ1$^+$ γδ T cell frequency compared to CVID patients with no genetic diagnosis (Fig. 4e). These findings confirm that known monogenic causes of CVID are not determinants of Vδ1$^+$ γδ T cell expansion, and that CMV viremia/disease independently has a significant impact on the frequency and phenotype of γδ T cell subsets in CVID patients.

**Age, CVID and CMV associate with a clonal Vδ1 TCR repertoire**

We next examined the γδ TCR repertoire to understand how CVID and CMV viremia may impact the clonality of Vδ1$^+$ and Vδ2$^+$ γδ T cells. To also examine the effect of aging on the repertoire clonality, we selected a cross section of different ages in both the healthy and CVID cohorts for repertoire analysis (see Supplementary Figs. 8 and 9 for full set of repertoire tree plots). Examination of the Vδ1$^+$ TCRδ repertoire revealed that the diversity index had a significant negative correlation with age in healthy individuals (Figs. 5 and 6a). In CVID patients, Vδ1$^+$ TCRδ repertoire diversity index was overall lower across all ages, with no significant correlation with age. Similarly, the CMV/CVID cohort had low Vδ1$^+$ TCRδ repertoire diversity across all patients featuring large, predominantly

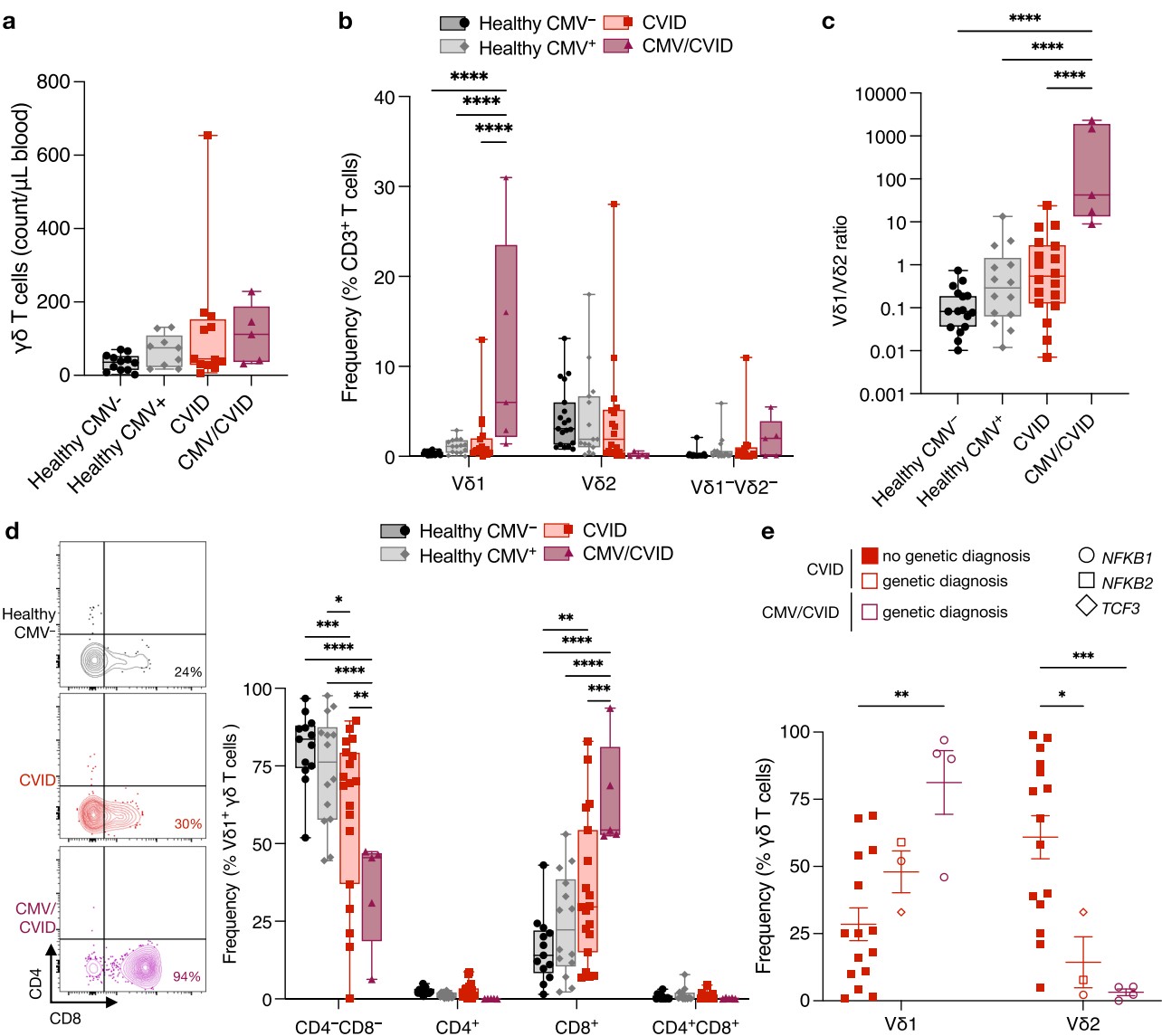

**Fig. 4 | Impact of CMV viremia on CVID patients γδ T cells.** PBMCs from healthy CMV seronegative individuals (CMV⁻), healthy seropositive individuals (CMV⁺), CVID patients, and CMV/CVID patients were analyzed by flow cytometry (see gating strategy in Supplementary Fig. 1). Graphs show (**a**) total γδ T cells counts (*n* = 11 CMV⁻, 9 CMV⁺, 12 CVID and 5 CMV/CVID), (**b**) Vδ1⁺, Vδ2⁺ and Vδ1⁻/Vδ2⁻ subsets as a frequency of total CD3⁺ T cells (*n* = 19 CMV⁻, 15 CMV⁺, 18 CVID and 5 CMV/CVID) ****$P < 0.0001$, and (**c**) ratio of Vδ1/Vδ2 (*n* = 21 CMV⁻, 14 CMV⁺, 18 CVID and 5 CMV/CVID) ****$P < 0.0001$. **d** Graph and plots show coreceptor (CD4/CD8) expression by Vδ1⁺ subset (*n* = 13 CMV⁻, 14 CMV⁺, 19 CVID and 5 CMV/CVID), where *$P < 0.05$ **$P < 0.01$ ***$P < 0.001$ ****$P < 0.0001$ (exact *P* values provided in source data file).

**e** Graph showing Vδ1⁺ and Vδ2⁺ subset frequencies separated based on no genetic diagnosis for CVID (*n* = 15) and genetic diagnosis for CVID (*n* = 3) and CMV/CVID (*n* = 4) cohorts, where *$P = 0.033$, **$P = 0.004$, ***$P = 0.0006$. For those with genetic diagnosis, symbol shape indicates specific gene implicated. For all graphs, each point represents an individual. For box and whisker graphs, line is at median, box is upper and lower quartiles, and error bars are minimum and maximum values. Statistical significance was calculated using either one-way ANOVA with Holm-Sidak's multiple comparison test with single pooled variance or two-way ANOVA with Sidak's multiple comparison test with single pooled variance. *NFKB1* nuclear factor kappa B subunit 1 gene, *NFKB2* nuclear factor kappa B subunit 2 gene, *TCF3* transcription factor 3 gene.

private, clonotypic expansions (Supplementary Table 5 and Fig. 5). Conversely, we did not observe a significant correlation with age or disease cohort in the Vδ2⁺ TCRδ repertoire diversity (Figs. 5 and 6a).

The composition of TCRδ and γ chain usage in the two γδ T cell subsets were generally consistent across cohorts. In the Vδ1⁺ population, TRDV1 was predominately paired with TRDJ1. Vδ1⁺ TCRγ usage was highly variable using genes TRGV2–TRGV9, predominately paired with TRGJ2 (Supplementary Fig. 10a). The only difference observed between cohorts was the use of TRGJ2, which was significantly lower in CVID patients (62 ± 7%) compared to healthy individuals (89 ± 2%). The Vδ2⁺ chains were consistent across cohorts, with the TCRγ clonotypes predominately using the TRGV9-JP chain and TRDV2 predominately paired with the TRDJ1 (Supplementary Fig. 10b).

Evaluation of shared clonotypes across samples revealed that there were no differences in overlap in the Vδ1⁺ repertoire across cohorts, with it being largely private (mean range = 15–23% clonotypes shared with at least one other individual) (Fig. 6b). When examining the TCRδ clonotype overlap between individuals, there was minimal overlap between any two individuals within the Vδ1⁺ TCRδ repertoire (1.4 ± 0.18%) with no clustering observed between different disease cohorts (Fig. 6c). The TCRγ repertoires for Vδ1⁺ were largely public (mean range = 78–92% clonotypes shared with at least on other individual) (Supplementary Fig. 11).

In contrast, the Vδ2⁺ TCRδ repertoire across samples was predominately public (mean range = 46–80% clonotypes shared) (Fig. 6b). The Vδ2⁺ TCRδ repertoire also had a higher overlap between any two

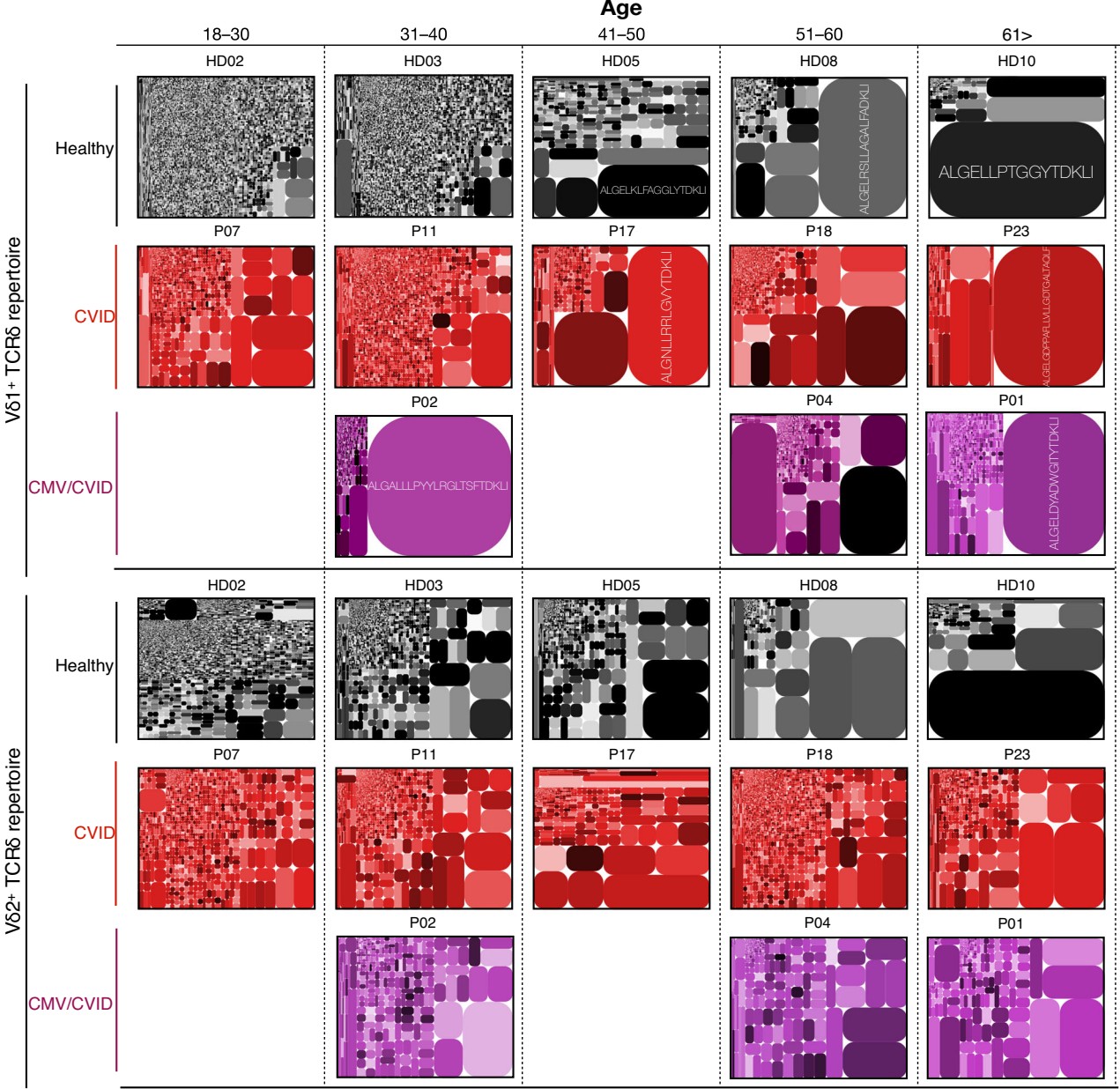

**Fig. 5 | TCRδ repertoire analysis.** Example TCRδ clonotype tree plots shown for Vδ1⁺ and Vδ2⁺ subsets from healthy individuals ($n = 10$), CVID patients ($n = 10$) and CMV/CVID patients ($n = 4$). See Supplementary Figs. 8 and 9 for full set of tree plots. Tree plots show unique clonotypes (box segments) as a proportion within the total repertoire (size of segment). Color of clonotype boxes do not match between plots or individuals. Major clonotypes have the CDR3δ sequence shown.

individuals ($5.5 \pm 0.36\%$), with results clustering based on individual samples rather than disease cohort (Fig. 6c). The TCRγ repertoires for Vδ2⁺ populations were largely public (mean = 85–96% clonotypes shared with at least on other individual) with the exception of the CMV/CVID cohort, which had a significantly more private Vδ2⁺ TCRγ repertoire ($58 \pm 14\%$) (Supplementary Fig. 11a).

These results collectively demonstrate that age, CVID, and CMV viremia all contribute to the clonal focusing of the Vδ1⁺ TCRδ repertoire, but these responses are largely private and do not correlate within healthy/disease cohorts.

### Vδ1⁺ TCR repertoire is stable over time in CMV/CVID patients

To examine the stability of the γδ TCR repertoire over time in CMV/CVID patients, we tracked the top TCRδ clonotypes from blood samples collected >1 year apart (see Fig. 3 for sampling timeline and

Supplementary Table 6 for patient infectious history during this period). We found that the Vδ1⁺ clonotypes remained stable over time, with >50% of the repertoire in all individuals comprising of maintained clonotypes present at both timepoints (Fig. 7 and Supplementary Fig. 12). In addition, the top clonotype for each donor consistently maintained its position as the dominant clonotype over time. In contrast, the Vδ2⁺ subset, which was only a minor population for all three CMV/CVID patients, exhibited a greater proportion of new and lost clonotypes over time compared to Vδ1⁺ subset. These findings reveal the long-term stability of the Vδ1⁺ subset clonality in CMV/CVID patients.

## Discussion

The factors that govern γδ T cell immunity are distinct to those that govern αβ T cells. This has been clearly demonstrated in studies of

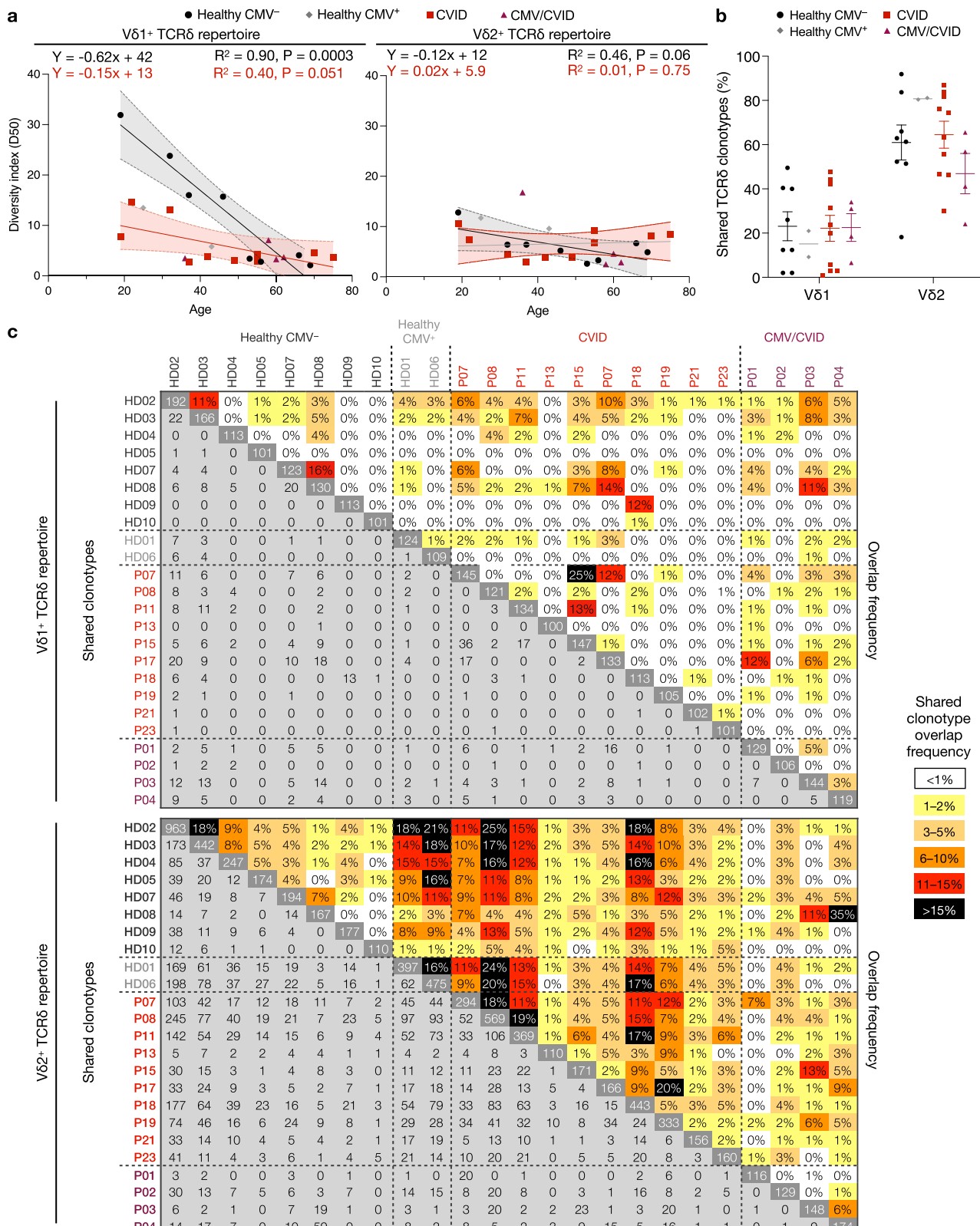

**Fig. 6 | TCRδ repertoire diversity and clonotype tracking.** TCRδ repertoire analysis for Vδ1⁺ and Vδ2⁺ subsets from healthy CMV⁻ individuals (*n* = 8), healthy CMV⁺ (*n* = 2), CVID patients (*n* = 10) and CMV/CVID patients (*n* = 4). **a** TCRδ diversity index for Vδ1⁺ and Vδ2⁺ subsets graphed versus age. Lines represent simple linear regression with 95% confidence interval of the best fit line shown as dashed line and shaded area for the healthy CMV⁻ and CVID cohorts. **b** Frequency of shared (present in at least one other sample) CDR3δ (amino acid) sequences, where line is at mean and error bars are SEM. **c** Number and frequency of TCRδ clonotypes (where abundance >50 sequencing reads). Plot shows number of shared clonotypes (light gray), frequency of shared clonotypes (color key), and total number of clonotypes for each sample (dark gray diagonal). TCR, T cell receptor.

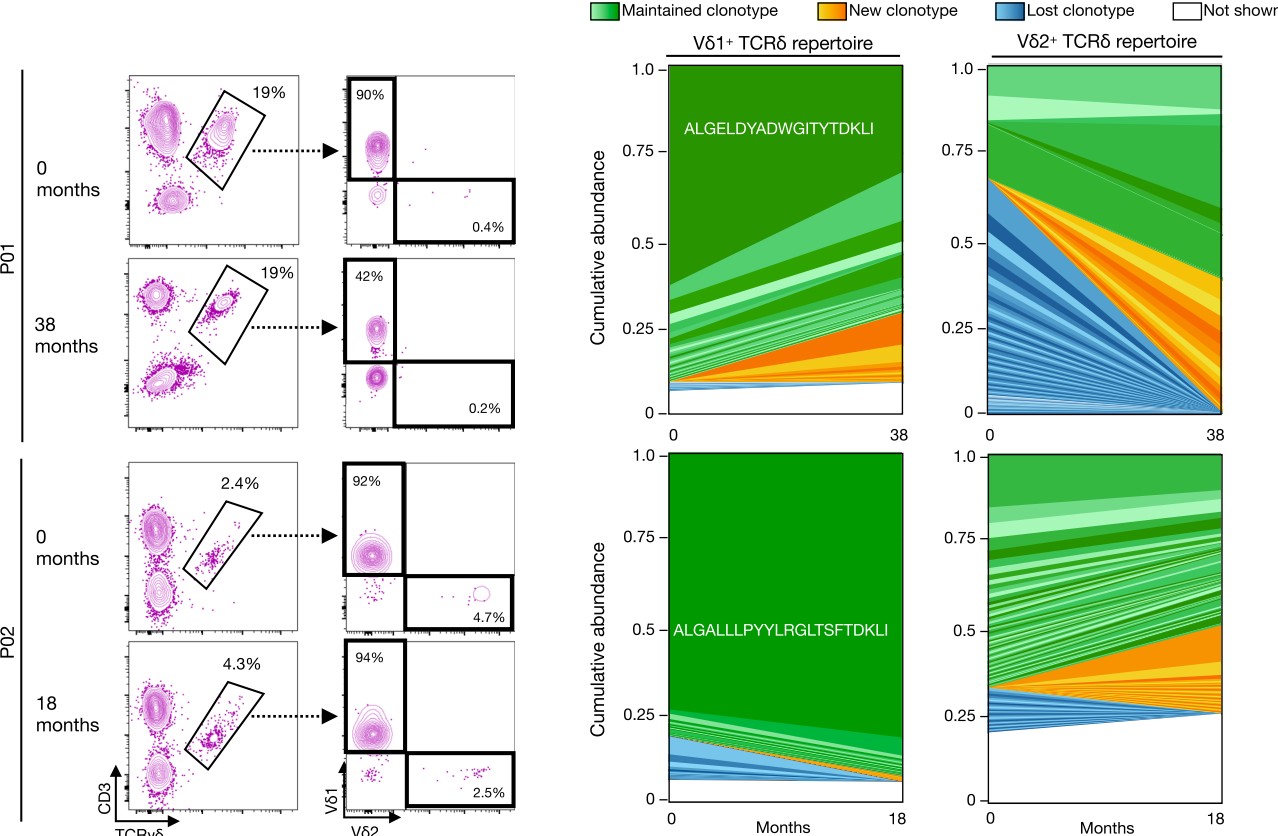

**Fig. 7 | TCRδ repertoire stability over time in CMV/CVID patients.** Plots show gating and frequencies of γδ T cells (as a proportion of CD3⁺ T cells) and γδ T cell subsets in two CMV/CVID donors, and corresponding shared clonotype abundance plots for Vδ1⁺ and Vδ2⁺ TCRδ repertoire showing the top maintained, new, and lost clones across 18 (P02) or 38 months (P01). Dominant clonotype TCRδ CDR3 sequence is shown. TCR, T cell receptor.

primary immunodeficiencies, where inborn errors of immunity that alter αβ T cells often do not influence or alter γδ T cells. In certain cases, the γδ T cells even expand to become the predominant T cell population[28–32]. Understanding the unique requirements and driving factors that govern γδ T cell biology enables a better understanding of their role in host immunity.

We found that, despite dysfunctional humoral immunity in CVID, the γδ T cell subsets are present and fully functional. γδ T cells have been shown to contribute to effective humoral immune responses[33–35], but there is limited evidence regarding whether γδ T cells rely on B cells for any aspects of their functionality[36]. In CVID patients, the Vδ1⁺ population displayed the expected effector and cytotoxic profile as well as clonotypic expansions of largely private repertoires. The Vδ2⁺ cells were mostly a minor population in CVID, but they still displayed a cytotoxic profile, and innate-like public repertoires that lacked clonotypic focusing. Both Vδ1⁺ and Vδ2⁺ subsets also exhibited intact and comparable responses to ex vivo stimulation to those from healthy individuals. Thus, our findings provide evidence that the γδ T cells do not require intact humoral immune responses as they are present and functional in CVID, and capable of participating in the host immune response against the patients' frequent infections.

Our results demonstrate the importance of considering age when examining the γδ T cell compartment in both health and disease. Increased γδ TCR clonality and a Vδ1/Vδ2 ratio inversion have been previously described in association with healthy aging[37,38]. The inversion of the Vδ1/Vδ2 ratio in our CVID cohort mirrors what we observed in our healthy aging cohort. Similarly, the reduced Vδ1⁺ TCRδ diversity across all ages in our CVID cohort was also detected in our aging healthy cohort. The extensive history of infections and inflammation, continuous antigenic challenges, and inflammatory microenvironment

in CVID patients may all contribute to the observed early onset and accelerated immune aging of γδ T cells in this disease setting.

We found that the Vδ1⁺ γδ T cells in CVID patients with CMV viremia (CMV/CVID) were skewed towards being CD8⁺ as well as being more expanded, activated, and cytotoxic compared to healthy individuals and CVID patients. This suggests that the Vδ1⁺ γδ T cells in CVID patients are still capable of mounting a robust adaptive-like response to CMV in the absence of normal B cell immunity. The γδ T cell response to CMV infection we observed in CVID patients aligns with previous reports in the immunocompromised settings of transplantation and congenital/infant infection. Across transplantation, neonatal infection, and now CVID, it is the Vδ1⁺ γδ T cells that become activated and expand in response to CMV infection[7,10,39–41]. This CMV-driven expansion of the Vδ1⁺ subset also occurs in healthy individuals, as supported by previous observations that CMV seropositive individuals have an elevated Vδ1⁺ subset frequency compared to seronegative individuals[9]. We observed a similar trend in our seropositive healthy cohort, but the increase did not reach statistical significance, likely due to our smaller sample size.

Our results demonstrate that the CMV-driven response of Vδ1⁺ γδ T cells is not impacted by the loss of NF-κB-mediated immune signaling. Notably, four of our five CMV/CVID patients had pathogenic mutations in *NFKB1* resulting in haploinsufficiency of the NFκB1 subunit p50, which is the most common inborn error underlying CVID[42–44]. Thus, we provide evidence that the normal Vδ1⁺ γδ T cell response to CMV infection remains unaffected, despite the loss-of-function in this key immune signaling pathway. This finding is consistent with the very limited investigations on NF-κB signaling in γδ T cells. These studies reported that loss of RelB, p52 (NF-κB2) or NF-κB-inducing kinase (NIK) from the NF-κB signaling pathway affects

interleukin (IL)−17A production by murine γδ T cells, whereas loss of p50 did not alter γδ T cell functionality[45,46].

Our analysis of the γδ TCR repertoire suggests that the Vδ1+ subset engages in an adaptive antigen-driven response to CMV, supporting the findings of previous studies[13,16,47]. In the fetal and young infant setting, where the majority of the TCR repertoire is public, a germline-encoded CDR3 Vγ8/Vδ1 population mounts a clonotypic antigen-driven response to CMV infection[41]. However, in the adult setting, where the TCRδ repertoires have become largely private, the CMV-induced clonotypic expansions observed are unique and non-overlapping between individuals[18]. This private adaptive response to CMV is also evident in our CMV/CVID cohort. Their clonal repertoire remained stable over the years of persistent CMV viremia, despite their many other complex and serious infectious complications during this period. However, it is still unclear whether these Vδ1+ clonal expansions confer the potential for memory-like responses to CMV.

The specific CMV antigens driving clonal responses of Vδ1+ γδ T cells are not well understood. They could be CMV-encoded antigens, CMV-induced stress-related antigens, or a combination of both. A previous report identified a CMV-induced expanded Vδ2− clone which was specific for endothelial protein C receptor (EPCR)[48]. A more recent study identified a Vδ1+ clone that recognized human leukocyte antigen (HLA)-DR in a peptide-independent manner[47]. Both these findings suggest that the ligands are CMV-induced stress-related antigens. These stress-related antigens could also potentially extend to the host lipidome, as CMV has been shown to induce stress-related alterations to the lipidome that may be detectable via CD1 lipid antigen presentation to γδ T cells[49–51]. However, further investigation is needed to understand both the nature and breadth of γδ T cell CMV antigens.

The clinical outcomes for our CMV/CVID cohort varied widely. Two patients that had a high burden of autoimmune dysregulation and major iatrogenic immunosuppression (prior to onset of CMV infection) experienced recurrent CMV disease that was indirectly associated with early mortality. One patient achieved a stable CMV viral load and resolution of end-organ disease following antiviral treatment. Two patients only required active surveillance due to their stable, low CMV viral loads and an absence of tissue-invasive disease. It is not known why some CVID patients are able to control CMV infection, while others progress to CMV disease. Further investigation into this is required to inform and guide CMV surveillance and early intervention strategies in CVID.

All the CMV/CVID patients in this study failed to clear the infection, despite having an expansion of Vδ1+ γδ T cells. Rapid expansion of circulating γδ T cells following renal transplantation in response to CMV reactivation has been associated with resolution of CMV infection and disease[52]. Thus, prospective systematic screening studies for CMV in CVID patients with long-term follow-up of individuals with CMV viremia (with or without progression to tissue-invasive disease) are needed to elucidate the immunological basis of CMV immune escape in CVID.

The absence of guidelines to screen and monitor CMV viremia/disease in CVID (at odds with the vigilance recommended post-transplantation) results in late detection and delayed treatment. This is likely a key factor in CMV-associated mortality in CVID[5,8]. Routine measurement of CMV-specific Ig is unreliable in the context of CVID, due to impaired antibody responses resulting in undetectable seroconversion, as well as administration of Ig replacement therapy. Regular testing for CMV DNA in peripheral blood would be a more appropriate test but is not currently routine clinical practice for CVID patients. This limited access and implementation of CMV exposure screening in CVID also prevented our ability to separate the CVID patient cohort into those that had never been exposed to CMV and those who were exposed but achieved viral load control with acute infection and no subsequent viremia.

CMV monitoring in the transplantation setting enables early or pre-emptive use of anti-viral therapies in the peri-transplant period. Even with implementation of CMV screening in CVID, outcomes may still be poor as there are major barriers to the use of anti-viral therapy for CMV in CVID patients. This is due to their high rates of baseline co-morbidities that preclude commonly used therapies. We have shown that CMV/CVID patients have an expanded Vδ1+ subset with their cytotoxicity and cytokine polyfunctionality intact. Thus, these Vδ1+ cells have the potential to aid in controlling CMV viral replication. Therefore, future development of targeted therapies directed at enhancing Vδ1+ T cells anti-CMV response, in combination with CMV screening, could lead to more favorable clinical outcomes for CVID patients.

Our study is limited by the small sample size of CVID patients that are diagnosed with CMV. In addition, unlike CMV reactivation during transplantation where a baseline/pre-transplant sample can be obtained, we were unable to obtain pre-CMV infection samples due to the unknown timing of when these individuals became infected. Thus, we cannot conduct longitudinal pre- and post-CMV infection analysis which would enable us to study γδ T cell kinetics during CMV infection. We are also somewhat limited in our conclusions due to the heterogeneous nature of CVID, with the complex infection history and varied genetic causes all potentially influencing the γδ T cells. The CMV antigens that drive the Vδ1+ subset expansion in CMV/CVID patients also remains elusive. However, despite these limitations our careful consideration of confounding factors such as age, sex, non-inflammatory complications, and CMV infection status have revealed key drivers of γδ T cell biology in CVID.

## Methods

### Study design

This study included individuals under the care of The Royal Melbourne Hospital (RMH) Clinical Immunology & Allergy Unit from 2016–2023 with participants identified via internal auditing[8]. CVID diagnosis was confirmed according to European Society for Immunodeficiencies (ESID) criteria[3] and medical history determined by reviewing medical records. CMV diagnosis was made based on positive CMV polymerase chain reaction (PCR) or immunohistochemistry on tissue samples (CMV disease) or from screening CMV viral loads performed on peripheral blood (CMV viremia) (Abbott RealTime PCR assay) (see Supplementary Table 2 for full CMV/CVID patient details). Patients who accepted the option for genetic testing were tested and diagnosed either through the Invitae platform (Primary Immunodeficiency multi-gene panel) or via recruitment into the Melbourne Genomics Health Alliance for genomic sequencing. Healthy donors were recruited through the volunteer blood donor registry at the Walter and Eliza Hall Institute of Medical Research (WEHI). Clinical data and sample tracking information was stored on a REDCap database (Vanderbilt University, v14.1.5).

Healthy donor age, sex, and medical history was self-reported and selection for inclusion in the healthy cohort was based on obtaining a broad range and balance of ages and sexes across individuals with no known inflammatory conditions. Ethical approval for this study was granted by the Human Research Ethics Committees of Melbourne Health (project ID: 2009.162) and WEHI (project ID: 10/02). Written, informed consent was obtained from all participants, in accordance with the Declaration of Helsinki prior to their participation in the study. Sample sizes were limited by the availability of eligible patients and healthy donors, but adequate to detect statistically significant differences between sample groups.

### Human sample processing

Blood samples were collected via venipuncture, whole blood was analyzed on an ADVIA 2120i hematology system to determine absolute cell numbers. PBMCs and plasma were isolated by density gradient

centrifugation using Ficoll-Paque Plus (Cytiva). Cells were cryopreserved in liquid nitrogen and plasma was stored at −20 °C.

## CMV serotyping

Plasma samples from healthy individuals were thawed and serotyped using an anti-CMV IgG human enzyme-linked immunosorbent assay (ELISA) kit (ab108724, Abcam) following manufacturer's instructions. Samples were considered seropositive when absorbance values were 10% above the cut-off control.

## Antibodies and staining reagents

All antibodies used were commercially available and validated for specificity by the manufacturer. Allophycocyanin (APC) anti-human IL-2 (MQ1-17H12, 1:20) and BD Horizon™ BV786 anti-human IFN-γ (4S.B3, 1:20) were purchased from BD Biosciences. Alexa Fluor 700 anti-human CD28 (CD28.2, 1:20), APC anti-human CD161 (HP-3G10, 1:10), Brilliant Ultraviolet (BUV) 395 anti-human CD3 (SK7, 1:20), Brilliant Violet 605 anti-human TCR Vα7.2 (3C10, 1:10), Brilliant Violet 650 anti-human CX3CR1 (2A9-1, 1:20), Brilliant Violet 711 anti-human IL-17A (BL168 1:20), Brilliant Violet 711 anti-human perforin (dG9, 1:20), PE/Dazzle 594 anti-human/mouse granzyme B (QA16A02, 1:100), PE/Dazzle 594 anti-human TNF-α (MAb11, 1:100), PE/Cyanine7 anti-human CD4 (SK3, 1:20), and PerCP/Cyanine5.5 anti-human CD8 (SK1, 1:20) were purchased from BioLegend. APC-eFluor™ 780 anti-human CD27 (O323, 1:20) was purchased from eBioscience. Phycoerythrin (PE) anti-human TCR Vδ2 (123R3, 1:100), fluorescein isothiocyanate (FITC) anti-human TCRγ/δ (REA591, 1:10), and VioBlue anti-human TCR Vδ1 (REA173, 1:10) were purchased from Miltenyi Biotec. Dead cells were excluded using the viability dye Zombie Aqua™ Fixable Viability Kit (BioLegend) according to the manufacturer's instructions.

## Flow cytometry and cell sorting

For standard flow cytometry cell staining, cells were thawed and stained with viability dye for 10 min at room temperature, followed by antibodies diluted in phosphate buffered saline (PBS) with 10% fetal bovine serum (FBS) (Sigma) for 20 min on ice. Flow cytometry data was collected on a BD LSRFortessa™ X20 running FACSDiva software (BD, v8). Cell sorting was performed on BD FACSAria™ Fusion Cell Sorter. Data were analyzed using FlowJo™ software (BD, v10).

## Bacterial culture

*Escherichia coli* was grown (at 37 °C in Lysogeny broth (LB) in shaking incubator) overnight, then diluted 1:100 in LB and grown to log phase for 3 h. $OD_{600}$ was measured and used to determine concentration.

## γδ T cell activation assay

PBMCs were cultured in RPMI 1640 medium supplemented with 10% FBS and antibiotics (at 37 °C and 5% $CO_2$ in humidified incubator) and pulsed with a range of stimulants. For cytokine stimulation: 100 U/mL human IL-2 (Abcam), 200 ng/mL human IL-12 (Miltenyi Biotec) and 200 ng/mL human IL-18 (BioLegend) were pulsed in two doses and cultured 72 h apart, and brefeldin A (BFA) (Invitrogen) added after second dose, before culturing for 18 h. For bacterial stimulation: *E. coli* at multiplicity of infection (MOI) 50 was added to PBMCs, cultured for 1 h prior to adding 50 µg/mL gentamicin (Sigma) and BFA and cultured for 18 h. For antigen stimulation: 10 ng/mL of HMBPP (Cayman Chemicals) was added and cultured for 4 h before adding BFA and cultured for 18 h. For phytohemagglutinin (PHA) stimulation: PHA-L (eBioscience) was added at 1× working concentration and cultured for 1 h prior to addition of BFA and cultured for 18 h. Following stimulation, cells were collected and stained with viability dye for 10 min followed by surface antibodies for 20 min on ice. Cells were fixed and permeabilized using the eBioscience™ Intracellular Fixation & Permeabilization Buffer Set (Invitrogen). Intracellular antibodies were

then incubated for 45 min at room temperature diluted in permeabilization buffer.

## TCR repertoire analysis

Up to 100,000 live cells were bulk-sorted into RNA*later* (Sigma-Aldrich) (sorted cell numbers for each sample are provided in Supplementary Table 7). RNA was extracted using an RNAmicro kit (Qiagen) according to the manufacturer's instructions. The human TCRγ and TCRδ chain iR profile kits (iRepertoire Inc.) were used to perform amplicon rescued multiplex–PCR to generate CDR3 amplicon libraries for sequencing following the manufacturer's instructions. Sequencing was performed using an Illumina MiSeq system. Raw sequencing data files were analyzed using iRweb (iRepertoire Inc.) to assign CDR3 sequences, variable (V), diversity (D), and junction (J) gene usage, calculate diversity indexes, and plot tree maps.

## Statistical analysis

Statistical analysis was performed using Prism (GraphPad Software, v10). For single comparison analysis, two-tailed unpaired T tests were used. For multiple comparisons with one independent variable, one-way analysis of variance (ANOVA) with Holm-Sidak's multiple comparison test was used. For multiple comparisons with two independent variables, two-way ANOVA with Geisser-Greenhouse correction and Sidak's multiple comparison test was used. Multivariate linear regression was used to calculate the associations between multiple independent variables and γδ T cell subset frequency and the β estimates reported. For all analysis, a significant result was indicated when $P < 0.05$.

## Reporting summary

Further information on research design is available in the Nature Portfolio Reporting Summary linked to this article.

## Data availability

The γδ T cell TCR repertoire raw sequence data generated in this study have been deposited in the NCBI Sequence Read Archive (SRA) under the BioProject Accession Number: PRJNA1055292. Source data are provided with this paper.

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

## Acknowledgements

We sincerely thank all the study participants. We would like to acknowledge Sylvia Tsang, Maureen Forde, the RMH Day Medical Centre, and the Volunteer Blood Donor Registry (WEHI) for assistance with sample collection. We thank Hannah Gosby for assistance with sample curation. We acknowledge the use of the services and facilities of Australian Genome Research Facility and WEHI flow cytometry facility. We also acknowledge the Melbourne Genomics Health Alliance, supported by the Victorian Government and Alliance members, for performing genomic sequencing of patients included in this study. We also wish to thank our colleagues in the Victorian Infectious Diseases Service, and the Haematology, Gastroenterology, Rheumatology and Respiratory Units at RMH, who have provided invaluable assistance in clinical management of these complex cases. This research was supported by: The Jack Brockhoff Foundation Early Career Medical Research Grant JBF 4847-2021 (L.J.H.), Monash University, Faculty of Medicine, Nursing and Health Sciences Early Career Postdoctoral Fellowship (A.v.B.), NHMRC Investigator Grant 2007884 (L.J.H.), NHMRC Early Career Fellowship 1161521 (M.K.Y.), Sir Clive McPherson Family Fellowship, Rae Foundation (V.L.B.) and WEHI Scientific Excellence PhD Scholarship (S.C.).

## Author contributions

Conceptualization: V.L.B. and L.J.H.; Investigation: S.C., B.M., M.M., A.J.F., E.L. and L.J.H.; Formal analysis: A.v.B and L.J.H.; Resources: S.C., M.K.Y., J.G., N.A.G. and C.A.S.; Data curation: S.C., M.K.Y., J.G. and N.A.G.; Project administration: S.C. and L.J.H.; Funding acquisition: V.L.B. and L.J.H.; Supervision: V.L.B. and L.J.H.; Writing – original draft: S.C. and L.J.H.; Writing – review & editing: S.C., A.v.B., V.L.B. and L.J.H.

## Competing interests

V.L.B. has undertaken investigator-initiated research on behalf of Immunosis and CSL. The remaining authors declare no competing interests.
