## [Peer Review File · Nature Communications]

Cytomegalovirus drives V δ 1+ $\gamma\delta$ T cell expansion and clonality in common variable immunodeficiencyREVIEWER COMMENTS

Reviewer #1 (Remarks to the Author):

The study by Chan et al. investigated $\gamma\delta$ T cells in the context of CVID disease, a disease characterized by impaired B cell responses. Studying $\gamma\delta$ T cells in immunodeficient patients provides an excellent tool to elucidate on the biology of human $\gamma\delta$ T cells. From this perspective results of this study provide valuable data. Nevertheless, as it has been reported by many other groups it is well established that V δ 1 T cells respond to CMV infection/reactivation in immunocompromised patients. The main findings of this study are that $\gamma\delta$ T cells are not impacted in CVID, and are capable to respond to CMV viremia. I have following comments on the results:

1. The authors focused on characterizing $\gamma\delta$ T cells. To place findings in a general context, I recommend to also show results, namely FACS data, for CD4 and CD8 $\alpha\beta$ T cells. At least in the methods it is stated that anti-CD4 and anti-V α 7.2 antibodies were included, and frequencies of CD4 and CD8 $\gamma\delta$ T cells were examined. The analysis of CD8+ $\alpha\beta$ T cells is of particular interest, when studying CMV viremia. Have the authors also performed TCR-seq of $\alpha\beta$ T cells as it would be important to also elucidate on clonal expansion of $\alpha\beta$ T cells upon CMV?
2. The authors claim that age and CMV impact on $\gamma\delta$ T cell subset distribution in CVID, but only looked at various clinical parameters (gender, CMV, age, disease) separately (Fig. 2 -3, maybe also Fig. 4). It is important to perform a statistical analysis on the abundance of V δ 1 and V δ 2 T cells using a multivariate linear regression model, and examine the β -estimates for the respective parameters.
3. Does the phenotype of $\gamma\delta$ T cells change with age in Fig 2?
4. On page 17 the bioproject numbers is not indicated.
5. In Fig 1B the authors use log-scale for $\gamma\delta$ T cell frequencies. This is a bit confusing. Do the absolute numbers of V δ 1 T cells change in CVID?
6. For a number of panels within Figures the number of samples/individuals are not clearly visible. It would be important to always provide exact numbers of samples studied.

Reviewer #2 (Remarks to the Author):

This paper describes phenotypic and quantitative characteristics of $\gamma\delta$ T cells in the context of common variable immunodeficiency (CVID), which is a heterogeneous disease with low antibody levels. For some analyses, CVID patients were stratified by cytomegalo virus (CMV) viremia. One question that was raised was whether normal $\gamma\delta$ T cell function requires intact B cell immunity, which is an interesting question given the presence of Fc receptors on $\gamma\delta$ T cells.

The main findings were that CVID and CMV are positively correlated with V δ 1 expansion and cytotoxic phenotype. For CVID, this is a new finding. However, it has been described in several prominent papers that are cited in the current paper, that V δ 1 cells expand in response to CMV and show increased cytotoxic phenotype, so this aspect of the findings is not novel. The main conclusion of the current paper is that the $\gamma\delta$ T cell population is not limited in size or changed in phenotype due to impaired B cell immunity in CVID.

All experiments, including flow cytometry, are technically well performed, analyzed, and clearly reported. The patient cohort is well described. My enthusiasm for this work is limited because it is mostly descriptive and correlative and does not lead to new disease-related hypotheses or new insights into the function of $\gamma\delta$ T cells. For this reason, I do not consider this work of significance to the general field of immunology or even the narrow field of $\gamma\delta$ T cell biology.

Reviewer #3 (Remarks to the Author):

This is an extremely well written manuscript that describes the gdT cell profile in 5 patients with CMV viraemia and CVID. The work also describes findings in a wide range of healthy donors and CVID cohort.

The findings are complementary to many studies of gdT cells in CMV disease in other settings but are new in this situation. The fact that 4 of the 5 patients had an NFkB mutation is noteworthy (although not overinterpreted)

The findings show expansion of private clones of Vd1 T cells with CMV viraemia. The specificity of these is unknown.

The writing and presentation is very clear.

I would suggest some discussion of number vs percentage. As I see it, the number of Vd1 T cells in CMV CVID is not increased (this is put in supplementary - perhaps should be main figure). Perhaps surprising given the article? But the % is increased over healthy CMV- donors (not a great control). And the % of Vd1 increases. So does this mean there is lymphopenia in CMV CVID and the retained Vd1 pool starts to represent a higher % ?

I would suggest rephrasing of the statement "Our results suggest that any future treatments for CMV that specifically target Vδ1+ γδ T cells would likely be effective in CVID patients". Does this mean kill them off or enhance them? Also, if they are there and expanded, why is disease still there? Are the populations exhausted? Indeed, it was not clear if the functional properties of the Vd1 cells in CMV CVID was determined.

Point-by-Point Response to Reviewers' Comments

Reviewer #1

We are very pleased Reviewer 1 states: *“Studying $\gamma\delta$ T cells in immunodeficient patients provides an excellent tool to elucidate on the biology of human $\gamma\delta$ T cells. From this perspective results of this study provide valuable data”.*

Comment 1:

The authors focused on characterizing $\gamma\delta$ T cells. To place findings in a general context, I recommend to also show results, namely FACS data, for CD4 and CD8 $\alpha\beta$ T cells. At least in the methods it is stated that anti-CD4 and anti-Va7.2 antibodies were included, and frequencies of CD4 and CD8 $\gamma\delta$ T cells were examined. The analysis of CD8+ $\alpha\beta$ T cells is of particular interest, when studying CMV viremia. Have the authors also performed TCR-seq of ab T cells as it would be important to also elucidate on clonal expansion of ab T cells upon CMV?

Response:

We agree with Reviewer 1 that the inclusion of FACS data on the CD4 and CD8 T cell frequencies will provide important contextual insight into the results of the paper. Therefore, we have included the circulating absolute numbers of: total T cells as well as conventional CD4 and CD8 T cells (with the normal healthy range indicated) in **Supplementary Fig. 7b**, frequencies for each CVID patient can be found in **Supplementary Table 2** and we have added the following to the main text:

(page 7, line 157–159)

*“As all other T cell counts were not significantly different across cohorts and largely within the normal range (**Supplementary Fig. 7b**) this suggests the expansion of the $V\delta 1^+$ subset is at the expense of the $V\delta 2^+$ subset.”*

Supplementary Figure 7b:

b

We also agree that evaluation of the $\alpha\beta$ TCR repertoire would be of potential interest in this cohort in terms of their response to the CMV infection, however, we did not perform this analysis as it was outside the scope of this study.

Comment 2:

The authors claim that age and CMV impact on $\gamma\delta$ T cell subset distribution in CVID, but only looked at various clinical parameters (gender, CMV, age, disease) separately (Fig. 2 -3, maybe also Fig. 4). It is important to perform a statistical analysis on the abundance of V δ 1 and V δ 2 T cells using a multivariate linear regression model, and examine the β -estimates for the respective parameters.

Response:

We thank Reviewer 1 for this suggestion and agree inclusion of multivariate analysis strengthens our conclusions regarding correlates of V δ 1 and V δ 2 T cell frequencies. We have now included this analysis in **Supplementary Table 3** and added the following to the main text:

(page 6, line 136–138)

*“Multivariate analysis confirmed that age and CVID, but not sex, are both significantly associated with an increased V δ 1 frequency (**Supplementary Table 3**).”*

(page 7, line 159–161)

“This correlation between increased V δ 1⁺ cell frequency and CMV/CVID was confirmed as significant by multivariate analysis (Supplementary Table 3).”

Supplementary Table 3:

Vδ1 frequency (% γδ T cells)				
Independent variable	β estimate	Standard error	P value	P value summary
Diagnosis [CVID]	15.54	5.42	0.0057	**
Age	0.39	0.14	0.0061	**
CMV viremia [positive]	47.17	9.79	<0.0001	****
Sex [Male]	-1.52	4.81	0.7527	NS

Vδ2 frequency (% γδ T cells)				
Independent variable	β estimate	Standard error	P value	P value summary
Diagnosis [CVID]	8.93	6.07	0.1466	NS
Age	0.44	0.16	0.0067	**
CMV viremia [positive]	-56.53	10.96	<0.0001	****
Sex [Male]	4.58	5.38	0.3979	NS

NS, not significant.

Comment 3:

Does the phenotype of γδ T cells change with age in Fig 2?

Response:

We have now performed phenotype vs age analysis and included this additional analysis as **Fig. 2e-f** and added the following to the main text:

(page 6, line 132–135)

“We then investigated the impact of age on effector status of the γδ T cell subsets and found a significant positive correlation between CD27⁻CX3CR1⁺ Vd1⁺ γδ T cells in both healthy ($R^2 = 0.14$) and CVID patients ($R^2 = 0.20$) (Fig. 2e) but no correlation between CD27⁻CD28⁻ Vd2⁺ γδ T cells and age (Fig. 2f).”

Figure 2e-f

Comment 4:

On page 17 the bioproject numbers is not indicated.

Response:

This has now been updated with the Bioproject accession number: PRJNA1055292 (page 18, line 441–442).

Comment 5:

In Fig 1B the authors use log-scale for $\gamma\delta$ T cell frequencies. This is a bit confusing. Do the absolute numbers of V δ 1 T cells change in CVID?

Response:

To improve clarity, Fig. 1b, 4b, and Supplementary Fig. 5b log scales have all been converted to linear. The absolute counts for $\gamma\delta$ T cell subsets have also been included in Supplementary Fig. 2a and added the following to the main text:

(page 5, line 102–104)

“This resulted in a significantly higher V δ 1/V δ 2 ratio in CVID (Fig. 1c), however, the cell counts for the $\gamma\delta$ subsets were not significantly different between healthy and CVID (Supplementary Fig. 2a).”

Supplementary Figure 2a

Comment 6:

For a number of panels within Figures the number of samples/individuals are not clearly visible. It would be important to always provide exact numbers of samples studied.

Response:

We have now updated all figure legends to include exact n values for each graph (rather than a range given for all graphs in figure).

Reviewer #2

We are pleased Reviewer 2 states: ***“All experiments, including flow cytometry, are technically well performed, analyzed, and clearly reported. The patient cohort is well described.”***

Comment 1

My enthusiasm for this work is limited because it is mostly descriptive and correlative and does not lead to new disease-related hypotheses or new insights into the function of $\gamma\delta$ T cells. For this reason, I do not consider this work of significance to the general field of immunology or even the narrow field of $\gamma\delta$ T cell biology.

Response:

While we do extensively describe our cohort, we also conduct in-depth quantitative analysis of the $\gamma\delta$ T cell phenotype, activation (**Fig. 1, 2 and 4**) and TCR repertoire (**Fig. 5, 6 and 7**) in CVID. Additionally, we perform functional analysis on these cells upon *ex vivo* stimulation (**Fig. 1g**).

We believe our study does lead to new and valuable insights into the function of $\gamma\delta$ T cells. Notably, we reveal that $\gamma\delta$ T cells can function independently of humoral immunity and remain unaffected by insufficient NF- κ B signaling — a concept not previously explored in human immunity. Furthermore, our research yields significant findings that generate disease-related hypotheses in the context of CVID. To date, studies on $\gamma\delta$ T cells and CMV in humans have predominately focused on transplant or neonatal settings. Importantly, our study marks the first evidence of V δ 1⁺ $\gamma\delta$ T cells activation and expansion in response to persistent CMV viremia in CVID, thereby expanding our understanding beyond transplant or neonatal settings.

We also extend these insights into clinically relevant hypotheses, in our discussion (**page 13, line 335–338**) we highlight that the intact functionality of $\gamma\delta$ T cell in the context of CVID positions them as ideal candidates for future therapeutic interventions (such as $\gamma\delta$ T cell adoptive transfer or $\gamma\delta$ T cell targeted vaccination strategies). Given the severe and potentially fatal outcomes we observed for CVID patients with CMV disease, in our discussion we also emphasize the need for routine CMV screening and the development of targeted therapies for this at-risk patient cohort (**page 13, line 320–329**).

With these novel findings and insights, we are confident that our study will attract a broad readership interested in not only in $\gamma\delta$ T cell biology, but also in virology, immunodeficiency, and clinical immunology.

Reviewer #3

We were very pleased with the positive feedback from Reviewer 3, who states: **“This is an extremely well written manuscript that describes the gdT cell profile in 5 patients with CMV viraemia and CVID... The findings are complementary to many studies of gdT cells in CMV disease in other settings but are new in this situation.”**

Comment 1

I would suggest some discussion of number vs percentage. As I see it, the number of Vd1 T cells in CMV CVID is not increased (this is put in supplementary - perhaps should be main figure). Perhaps surprising given the article? But the % is increased over healthy CMV- donors (not a great control). And the % of Vd1 increases. So does this mean there is lymphopenia in CMV CVID and the retained Vd1 pool starts to represent a higher % ?

Response:

We agree with Reviewer 3's point regarding discussion on $\gamma\delta$ T cell numbers vs. percentage (also suggested by Reviewer 1). We have now moved the absolute number $\gamma\delta$ T cell graph from supplementary figure to the main figure (**Fig. 4a**). We have also included the circulating absolute numbers of total T cells as well as conventional CD4 and CD8 T cells (with the normal healthy range indicated) in **Supplementary Fig. 7b**, and added the following to the main text:

(page 7, line 151–159)

*“CMV/CVID patients had no significant difference in total circulating numbers of $\gamma\delta$ T cells compared to healthy and CVID cohorts (**Fig. 4a**) and only a significantly higher frequency of total $\gamma\delta$ T cells compared to CMV⁻ healthy individuals (**Supplementary Fig. 7a**). Despite this, we did observe a significant increase in the V δ 1⁺ subset frequency ($12 \pm 5.5\%$) within the T cell compartment compared to CMV⁻ and CMV⁺ healthy individuals and CVID patients (**Fig. 4b**). The V δ 1/V δ 2 ratio was also significantly more inverted in CMV/CVID patients (778 ± 481) compared to CVID patients (2.9 ± 1.4) (**Fig. 4c**). As all other T cell counts were not significantly different across cohorts and largely within the normal range (**Supplementary Fig. 7b**) this suggests the expansion of the V δ 1⁺ subset is at the expense of the V δ 2⁺ subset.”*

Figure 4a

Supplementary Figure 7b

Comment 2

I would suggest rephrasing of the statement "Our results suggest that any future treatments for CMV that specifically target V δ 1+ $\gamma\delta$ T cells would likely be effective in CVID patients". Does this mean kill them off or enhance them? Also, if they are there and expanded, why is disease still there? Are the populations exhausted? Indeed, it was not clear if the functional properties of the V δ 1 cells in CMV CVID was determined.

Response:

To enhance clarity regarding our discussion on the potential of anti-CMV $\gamma\delta$ T cell therapy we have now rephrased this paragraph of the discussion, which now reads:

(page 13, line 330–338)

"CMV monitoring in the transplantation setting enables early or pre-emptive use of anti-viral therapies in the peri-transplant period. Even with implementation of CMV screening in CVID, outcomes may still be poor as there are major barriers to the use of anti-viral therapy for CMV in CVID patients. This is due to their high rates of baseline co-morbidities that preclude commonly used therapies. We have shown that CMV/CVID patients have an expanded V δ 1⁺ subset with their cytotoxicity and cytokine polyfunctionality intact. Thus, these V δ 1⁺ cells have the potential to aid in controlling CMV viral replication. Therefore, future development of targeted therapies directed at enhancing V δ 1⁺ T cells anti-CMV response, in combination with CMV screening, could lead to more favourable clinical outcomes for CVID patients."

We agree with Reviewer 3 that clarification on the functional properties of the CMV/CVID patients V δ 1 cells is important, and have now included this analysis in **Supplementary Fig. 7e** and added the following to the main text:

(page 7, line 165–167)

*"We also confirmed that the V δ 1⁺ $\gamma\delta$ T cells still maintained a polyfunctional cytokine response to stimulation comparable to healthy and CVID cohorts (**Supplementary Fig. 7e**)."*

Supplementary Fig. 7e

REVIEWERS' COMMENTS

Reviewer #1 (Remarks to the Author):

In the revised manuscript version the authors answered all the raised concerns and included new supportive analysis. Albeit descriptive, the study provides insights into expansion potentials of virus-reactive gd T cells in patients with common variable immunodeficiency.

Reviewer #2 (Remarks to the Author):

I do appreciate the author's rebuttal to my point that this study does not lead to new insights in gd T cell biology or CVID. In response, they point to the new insight that gd T cells apparently function normally despite altered antibody concentrations and NK-kB signaling, which is true. In response to my remark that the study does not lead to new clinically relevant hypotheses they point to their idea that because of their intact function, gd T cells are ideal candidates for future therapy, which is not the type of incisive or novel hypothesis I am hoping to find in Nat Comms. I still feel a lack of enthusiasm for this work for the reasons I mentioned in my first review. It is my duty to point this out to the editors so it can be considered in their decision process.

Reviewer #3 (Remarks to the Author):

The authors have addressed the comments of the Reviewers well. The significance and clarity is now much more clear.
No further revisions are needed.